# Schwann Cell Cultures: Biology, Technology and Therapeutics

**DOI:** 10.3390/cells9081848

**Published:** 2020-08-06

**Authors:** Paula V. Monje

**Affiliations:** 1The Stark Neurosciences Research Institute, Indiana University School of Medicine, Indianapolis, IN 46202, USA; pmonje@iu.edu; Tel.: +1-317-278-9432; 2The Department of Neurological Surgery, Indiana University School of Medicine, Indianapolis, IN 46202, USA

**Keywords:** primary Schwann cell cultures, explant cultures, proliferation, differentiation, myelination, cell transplantation

## Abstract

Schwann cell (SC) cultures from experimental animals and human donors can be prepared using nearly any type of nerve at any stage of maturation to render stage- and patient-specific populations. Methods to isolate, purify, expand in number, and differentiate SCs from adult, postnatal and embryonic sources are efficient and reproducible as these have resulted from accumulated refinements introduced over many decades of work. Albeit some exceptions, SCs can be passaged extensively while maintaining their normal proliferation and differentiation controls. Due to their lineage commitment and strong resistance to tumorigenic transformation, SCs are safe for use in therapeutic approaches in the peripheral and central nervous systems. This review summarizes the evolution of work that led to the robust technologies used today in SC culturing along with the main features of the primary and expanded SCs that make them irreplaceable models to understand SC biology in health and disease. Traditional and emerging approaches in SC culture are discussed in light of their prospective applications. Lastly, some basic assumptions in vitro SC models are identified in an attempt to uncover the combined value of old and new trends in culture protocols and the cellular products that are derived.

## 1. Introduction

Schwann cells (SCs) are axon-ensheathing cells that naturally reside in the peripheral nervous system (PNS) of all vertebrate species. These glial cells derive from multipotent neural crest precursors that migrate within nascent peripheral nerves and take a long time to develop fully in response to a myriad of axonal and environmental cues [1,2]. Proliferative SC precursors (SCPs) sort individual axons while chaperoning their growth cones as they proceed through the developing peripheral nerve yet without providing ensheathment. They continue to mature during embryonic life to eventually form a basal lamina, cease proliferating, restrict their differentiation potential, and ultimately ensheath axons as dictated by the type of axon they engage with. SCs mature in two distinct conformations, namely by establishing a unitary relationship with an axon to originate a *myelinating* cell, or by enclosing multiple axons of lesser caliber to originate a *non-myelinating* cell, also referred to as the Remak SC. It is understood that, with the exception of embryonic SCPs, all SCs are unipotent, lineage-committed cells. SCPs are able to give rise to cell types other than SCs with different anatomical locations, including endoneurial fibroblasts, melanocytes, parasympathetic neurons, enteric neurons, chromaffin cells of adrenal glands, and even mesenchymal stromal cells (reviewed in [3]). 

One key feature of the biology of SCs is that their differentiation into myelinating and non-myelinating cells is not terminal but reversible. Indeed, SCs are among the few known somatic cell types in the adult mammalian body that respond to an injury by reprogramming their phenotype through dedifferentiation, self-renewal and conversion into repair cells capable to foster nerve growth [4]. SC dedifferentiation is a form of somatic cell reprogramming which endows the PNS with a unique self-repair capability that is absent in the central nervous system (CNS), [5].

For decades, the natural plasticity of the SC has been exploited to generate cultured cells alone and together with neuronal systems. The culturing of adult nerve-derived SCs is possible because SCs within a dissected nerve fragment or ganglia survive well after being isolated from the body. If such nerve fragment is placed in culture, the SCs within it do not die but instead, they recapitulate key events typically associated with injury-induction such as myelin engulfment and degradation, dedifferentiation, and cell division [5]. Notoriously, it has been shown that SCs from cadaveric human nerves not only survive for a long time under cold storage but also remain potent to generate expandable SC cultures if placed under appropriate in vitro conditions [6,7]. These advantages have provided researchers with an opportunity to transplant isolated SCs or nerve tissue grafts to foster neuroprotection, neuroregeneration and remyelination after trauma or disease (reviewed in [8,9,10]).

Isolated embryonic, postnatal, and adult PNS-resident SCs can proliferate extensively in vitro and give rise to relatively uniform adherent cultures that retain essential SC characteristics. Given the fact that myelinating and non-myelinating SCs are reprogrammed after injury and get activated even when the nerves are excised from the body, it is conceivable that any type of nerve (i.e., motor or sensory) would serve to derive primary cells. Current protocols for SC culturing have mostly used the largest nerve in the body, the sciatic nerve, because of practical rather than biological reasons. For instance, clinical protocols for human SC culture have used the sural nerve, a sensory branch from the sciatic nerve in the leg, because its surgical removal can be relatively well tolerated by the patient [8,9]. Cultured SCs display unambiguous features regardless of the nerve of origin and the age of the donor though SCs from immature nerves (or younger donors) have a tendency to proliferate at a faster rate [6]. The properties of SCs from primary and established cultures are rather constant and the cells are easily identifiable by microscopic observations despite the fact that some degree of heterogeneity exists in most SC populations and species-specific differences are revealed at the molecular level or in individual cells (reviewed in [11]). 

Currently, there are two essentially distinct approaches to obtain cultured SCs for experimentation: (1) direct isolation from a tissue enriched in lineage-committed SCs or their progenitors; and (2) artificial creation or induction by the hand of the experimenter based on a stem cell approach, directed differentiation, or conversion from somatic cells. The applications and limitations of man-made SCs, which are sometimes referred to as engineered or induced SCs (iSCs), have been described previously (reviewed in [12]). For that reason, this review aims to elaborate on classical approaches that comprise the isolation of SCs directly from PNS tissues and their establishment in culture. Furthermore, it aims to explain the major features of normal cultured SCs and their value as cells that are representative of, and can model, the SC phenotype in vitro and the sophisticated relationships that SCs establish with other cells and the environment. 

As opposed to adult CNS glia, SCs are ubiquitous and relatively accessible for experimentation. The methods for SC culture and differentiation are robust and have been optimized over many decades of systematic work in independent laboratories. With the proper handling, high numbers of quality grade, biologically competent SCs can be obtained from developing and mature PNS tissues. For instance, SCs can be isolated from assorted mouse models [13] and transgenic animals expressing gene reporters. In this way, researchers have been able to compare cell behavior in vitro and in vivo, and trace the fate of transplanted SCs to grasp the associated mechanisms of repair, migration and myelination in central and peripheral nerve lesions [10,14,15,16]. A particular advantage of SCs is that it is possible to manufacture them at a large scale from human autologous tissues following guidelines compatible with safe manufacturing practices, which has made it possible to re-implant cultured SCs for therapeutic benefit in human patients [17]. Another important advantage of cultured SCs is the possibility to achieve full SC differentiation (myelination) with an organotypic organization. Myelin formation by cultured SCs can be both achieved in ganglionic (explant) cultures and cocultures of graded complexities generated by means of recombining SCs and neurons, thus enabling accurate modeling of the myelination process in vitro. 

The range and variety of technologies that have used cultured SCs is overly extensive and cannot be covered fully in this review. The notion that SCs can be grown and differentiated in vitro dates from the 1950s with cytological investigations of cells in tissue culture [18]. For this reason, and by way of introduction to the culturing of SCs, the evolution in culture technologies over the past 60 years has been summarized in the section below. The ultimate goal is to help researchers realize what the value and limitations of nerve-derived SC cultures are, and how much potential exists for using them as biotherapeutic tools and reliable models for neurodevelopmental processes, trauma, and neurodegenerative disease. 

## 2. The Evolution of Schwann Cell Cultures: A Three-Tier Story at a Glance

The first unequivocal evidence of the SC phenotype in vitro can be traced to the study of Peterson and Murray (1955), reporting the appearance of PNS- or SC-specific myelin in long-term living cultures of chick embryo spinal ganglia [19]. A decade after this discovery, organized explant cultures were established from fetal rat dorsal root ganglia (DRG) with the key contributions of Mary and Richard Bunge [20]. The authors of this study performed a meticulous analysis of the rat explant cultures at the levels of both light and transmission electron microscopy (TEM). Their analysis revealed that these cultures consisted of an organotypic model of sensory ganglion tissue encompassing myelinated and unmyelinated fibers surrounded by endoneurial and perineurial components. Since then, ganglionic DRG cultures have proven instrumental to understand normal and pathological SC–neuron relationships. These cultures offer the advantage of being relatively easy to establish and maintain without the need to introduce special substrates, media, or intermediate steps with potential to alter the properties of resident neurons and glial cells. To date, they represent the most faithful organized system able to recapitulate the complexity of the axon-SC unit, the anatomy of myelinating cells, and their spatial organization in fascicles surrounded by connective tissue layers [18], (Figure 1 and Figure 2).

The need to avoid ambiguities caused by the impact of fibroblast, and to a lesser degree macrophage, overgrowth in DRG explant cultures, motivated efforts in the 1970s to reconstitute myelinating SC–neuron systems from purified cellular components [21]. By recombining isolated SCs and DRG neurons for the first time, it was demonstrated that the axons of purified DRG neurons were sufficient to activate SC proliferation by means of expressing a neural mitogen that was located on the axonal surface [22,23,24,25]. Later studies showed that whereas neuronal cells could support the proliferation and differentiation of SCs, and the assembly of a basal lamina on the abaxonal SC surface, nerve-plus-SC cultures were not able to originate perineurium or the larger diameter collagen fibers proper of the endoneurial matrix in the absence of other cell types [26]. 

Since then, many systematic implementations contributed to the modernization and standardization of the protocols we use today to obtain and expand SC cultures divested of neurons. Setting these standards required the identification of simplified substrates and soluble growth factors for efficient SC propagation, and the discovery of selective markers to characterize and purify SCs from mixed cultures [27,28,29]. Tools such as antibodies and assays for myelin markers had to be developed to understand the stages of SC maturation [30,31,32]. One key finding was the discovery of members of the neuregulin family of growth factors to be mitogenic for cultured SCs [29,33,34], and their role in mediating axon contact-driven SC proliferation [35]. Another key finding was the observation that provision of agents inducing cyclic adenosine monophosphate (cAMP) promoted maximal SC proliferation in the absence of neurons [36,37,38]. The identification and purification of GGF (also known as ARIA and Neu differentiating factor or NDF), [29,39], which is a member of the neuregulin family, and its combination with agents that elevate intracellular cAMP, a synergistic adjuvant of GGF-dependent proliferation, inspired the formulation of culture media supportive of SC growth and the preparation of SC cultures under defined conditions. 

The history and milestones in the progression of SC culture technologies are represented in Figure 1. These technologies evolved mainly during the 1970s through the 1990s when intensive investigations were conducted to achieve SC expansion without neurons or transformation. Further investigations since the mid 2000s, which are still in progress, have had to goal to achieve the SC phenotype itself without procuring it from PNS tissues. A new era in cell culture technologies with potential to generate patient-specific human SCs has emerged with the capability to derive the SC phenotype in the laboratory from non-neural cells. To approach these transitions, the narrative was divided in three steps explaining key basic technological and conceptual innovations regarding the preparation of neuron-based (Section 2.1), neuron-free (Section 2.2), and iSC cultures (Section 2.3). Investigations on the derivation of immortalized SC lines date from the 1970s [40] but their value in understanding SC physiology is nowadays regarded limited. A few rodent SC lines are available and have been used in various basic studies but they are not deemed necessary considering the ease of culturing of primary SCs and their propagation in the long term without apparent loss in phenotype. 

### 2.1. SCs with Neurons and the Dependency on Axon Contact for SC Expansion and Differentiation

For many years, researchers could not achieve an efficient propagation of SCs in the absence of neuronal cells. The only way known to attain expansion and purification of the SC populations was to serially transfer DRG ganglia from and onto collagen-coated surfaces and collect the cells that emerged from the outgrowth of neurites when the ganglia containing the neuronal bodies were excised [18]. As explained above, it was early noted that axons provide both the scaffolding and the chemical signals needed for SC adhesion and mitosis. Evidence from early empirical observations showed that SCs stopped proliferating when deprived of neuronal support and that the cells remained alive but quiescent until contact with axons was restituted [22]. Researchers understood that SC alignment to axons and direct apposition between the axolemma and the SC membrane were necessary for axon-induced SC mitogenesis. What was also understood was that the mitogenic activity was membrane-bound [24] and could be recovered in isolated membrane fractions from the neuronal cultures [23] but the identity of such mitogenic activity was unknown and at the time, this limited the possibility to significantly amplify the SC cultures without the contribution of the neuronal component (Section 2.2). 

For many years, the ganglionic myelinating explant cultures were the only available system in which SC differentiation could be achieved (Figure 2). By the 1970s, it was known that proper adhesion to an extracellular matrix (ECM) was required for SC–axon alignment and myelination [41] but factors in the culture medium itself also played a role. When the neuron–SC cultures were maintained in a traditional complex medium containing serum and embryo-extract, the SCs adhered to, and aligned with, the axons as they grew out from the explants, the characteristic SC basal lamina was formed, and abundant myelination occurred. However, these hallmarks of organotypic SC differentiation did not occur in chemically defined medium, suggesting that key ingredients present in serum and/or embryo extract were needed for SCs to complete ensheathment and deposit an ECM prior to completing their differentiation into myelinating cells [42]. The feasibility to establish sustainable myelinating cultures consisting only of normal SCs and ganglionic (non-dissociated, purified) DRG neurons was demonstrated in the mid-1970s by the study of Patrick Wood [21]. The recombination of glia and neuronal components illustrated that SCs could myelinate axons without the contribution of other cell types such as fibroblasts [21] but the culture system had to be refined over the ensuing decade to enhance neuronal purity and increase the extent of myelination that could be achieved. The improved myelination system, even used today, included steps to enzymatically dissociate the DRG ganglia into single cells and purify them neatly prior to the addition of the SCs [43]. Further experiments with the purified DRG neuron–SC culture system proved especially enlightening with regard to the cellular and molecular control of SC differentiation. The discovery of L-ascorbic acid (vitamin C) as active ingredient required for SC myelination, and the understanding of its mechanisms of action in supporting ECM deposition and basal lamina formation, opened doors to manipulating the extracellular conditions to support the full differentiation of SCs in co-culture with DRG neurons [43,44]. 

Despite that SCs can proliferate in response to neurites originating from cell lines such as PC12 cells [45], to date, replacing the DRG neurons in the form of ganglionic explants or dissociated neuronal cells as substrate supportive of myelination has been challenging. Since the early 1980s investigators have known that, as opposed to oligodendrocytes, SCs (rat or human) placed in culture autonomously downregulate the expression of myelin-associated proteins and glycolipids unless induced to myelinate [30,31,32]. Even though SCs in monoculture can enhance the expression of myelin markers with provision of exogenous inductive signals (reviewed in [46]), continued stimulation from appropriate axons is required for further maturation and myelinogenesis. 

### 2.2. SCs without Neurons and the Advent of Soluble Mitogenic Factors 

The preparation of expandable SC cultures in isolation from neurons was first reported in the study by Raff et al. (1978), [37]. This group used culture media that contained a crude extract of bovine pituitary gland combined with cholera toxin, a strong inducer of cAMP, to enhance the proliferation of neonatal rat sciatic nerve SCs [36]. Furthermore, in a separate study they devised an efficient method to selectively label and eliminate the rapidly dividing fibroblast populations. The removal of fibroblasts was achieved by using complement-mediated cell lysis of Thy-1-positive cells [50] after showing that antibodies against the cell surface protein Thy-1 selectively identified fibroblasts rather than SCs in live cell cultures. With these tools, these researchers prepared cultures consisting of >99% pure SCs that could be amplified for 6 consecutive passages in purified form. The expanded SCs retained high levels of expression of cell surface Ran-1 (217c), an antigen which matched the low-affinity nerve growth factor (NGF) receptor or NGFR [51], and intracellular S100β, which is a calcium binding protein widely used as a SC marker (Section 6.1). Altogether, these studies paved the way to optimizing the protocols for the culturing, purification and immunological identification of neuron-free primary SCs from humans [52,53] and many other species. The protocol by Brockes et al. [50], or an adapted version of it, continues to be used today by the scientific community worldwide. 

An important contribution from the above-mentioned studies was the finding that an agent within the pituitary extract potently induced SC proliferation. This agent was present in brain extract, albeit to a lesser extent, but it was not present in serum and it did not promote cAMP elevation. However, the combination of pituitary extract and cholera toxin synergistically induced SC proliferation [36]. This discovery ratified that it was possible to effectively propagate SCs in the presence of soluble components and in the absence of neurons. It was the prelude to a series of efforts that led to the identification of a family of proteins collectively referred to as neuregulins to be mitogenic for SCs. The neuregulins are ligands for the ErbB/HER family of membrane tyrosine kinase receptors whose first described representative was the epidermal growth factor receptor (EGFR/ERbB1). Mechanistically, the identification of a neuregulin-like activity and subsequently, the purification of the protein, allowed to explain why direct axon-to-SC signaling by contact drives SC division. On one hand, it was found that sensory axons expressed a membrane-bound form of neuregulin. On the other hand, it was found that SCs expressed two types of hetero-dimerizing ErbB receptors, namely ErbB2 and ErbB3, whose activation was necessary for the proliferation of rat SCPs [54] and adult human SCs [35,55]. Because maximal levels of SC proliferation are seen when neuregulin and cAMP-inducing agents are provided in combination, it was suggested that together these two factors mimic the strong promitogenic effect of axons. Whereas the identification of the adhesion GPCR (GPR126) has provided evidence on the mechanistic linkage between cAMP signaling and myelination in SCs [1], an analogous ligand-receptor system responsible for axon contact-mediated SC proliferation via cAMP is so far ill-defined. 

### 2.3. iSCs and the Advent of Technologies to Recreate the SC Phenotype In Vitro

The concept of *SC creation* is rather new [56]. In this paradigm, cells with SC characteristics (iSCs) are derived by using directed in vitro differentiation (e.g., from multipotent neural or non-neural stem cells) [57], transdifferentiation [56] or even direct conversion from somatic cells such as fibroblasts [49,58]. iSCs can be generated from embryonic stem cells, induced pluripotent stem cells (iPSCs), [59], neural stem cells (e.g., neural crest-derived) and mesenchymal stem cells (MSCs) originating from the bone marrow [60,61,62] or adipose tissue [63]. Another source is the skin, which contains at least two independent multipotent dermally derived precursors of neural crest [57] and mesenchymal origin [64] able to convert into iSCs in vitro. 

iSCs are usually the end result of a number of orderly steps of induction using chemicals added to the culture medium (e.g., small molecules and growth factors) alone or together with genetic manipulation of the cells. The number of steps, and the way they are instrumented, may be determined by the cell of origin. Ultimately, most methods rely on treatment with neuregulin, alone or together with a cAMP-inducing agents, to drive differentiation along the SC lineage. Regarding genetic approaches, it has been shown recently that the enforced expression of two transcription factors with SC specificity (Erg2/Krox20 and Sox10) suffices to convert human and rodent skin fibroblasts into SC-like cells with capacity to wrap axons and generate compact myelin [65].

iSCs most often display high levels of genes typically found in neural crest cells and/or SCPs, such as Sox10, NGFR, glial fibrillary acidic protein (GFAP), peripheral myelin protein-22 (PMP22), and proteolipid protein-1 (PLP1), [3]. The expression of transcripts encoding for neurotrophins such as NGF, brain-derived neurotrophic factor (BDNF), and glial cell-derived neurotrophic factor (GDNF) has also been confirmed [59] along with the expression of S100β and Egr2. In recent years, researchers have taken advantage of high resolution omics approaches such as RNAseq to identify iSCs from normal and diseased patients [65]. In addition, they have used myelination and neurite outgrowth assays, and transplantation of iSCs in CNS or PNS lesions, to assess their properties in vitro and in vivo, respectively. Nevertheless, the myelination efficiency of iSCs is often low and the populations that are obtained are heterogeneous regarding their acquisition of SC traits [59,62]. 

These technologies are in progress but already have gathered major attention in the field. This is due mainly to the prospect of obtaining human SCs without major ethical concerns. SC derivation from iPSCs offer an alternative, potentially unlimited source of patient-specific human SCs with broad applications in modeling genetic disorders that affect SC function such as Charcot-Marie-Tooth disease [49]. Yet, how much iSCs resemble the SCs from peripheral nerve or how stable they are as they are passaged or manipulated in culture remains to be further explored.

## 3. Methods and Protocols 

Overall, SCs are a highly amenable phenotype for cell culture work once established in vitro. The first and most important limiting step of the culture workflow is the isolation of the mother SCs defined as those SCs used to initiate the primary culture. The type, quantity and quality of the mother SCs determine the attributes of the preparations that are produced. The high proliferative potential of most nerve- and ganglia-derived SC populations, in concert with their phenotypic and genetic stability, allows repetitive rounds of subculture in medium containing SC-specific growth factors (Figure 3). Once the mother cells are obtained and their adhesion to a substrate is secured, similar protocols can be used for maintenance, expansion and purification of the populations. Generally, primary SCs can be managed as if they were established cell lines for as long as the cells maintain their normal SC characteristics, as highlighted mainly by the anchorage and growth factor dependency of cell division. Standard interventions can be made to label the cells transiently or permanently, and infuse a transgene of interest via transfection or viral infection. Besides, it is possible to subject primary or expanded SCs to cryopreservation without affecting their biological activity. Albeit some exceptions, expanded SC cultures are fairly simple, which facilitates cell type-specific labeling, tracing, and purification to remove undesirable contaminating phenotypes. 

The literature on culture methods for primary SCs is far-reaching. This section is not meant to provide a thorough description but rather an overview to guide the experimenter on finding the best suited method to a particular experimental goal. How best to isolate and culture SCs depends on the characteristics of the starting material, the type, amount and quality of cells needed, and the downstream applications (Figure 3). Mass producing SC preparations for preclinical or clinical studies is feasible but certain applications requiring extended passaging can be limited [66]. 

The following sections have been organized to describe general concepts and considerations regarding the processing of tissues (Section 3.1.), the isolation of the mother SCs (Section 3.2.), and their amplification to generate small-, medium- or large-scale cultures (Section 3.3). Postculture steps, including the identification and purification of established SC populations (Section 3.4), their labeling (Section 3.5) and banking (Section 3.6), are also described. 

### 3.1. Choice and Procurement of Source Tissues 

The SC is a highly abundant cellular phenotype. Being the glial cells in the PNS, SCs spread throughout the body. In addition, cells with normal SC characteristics have been found in the spinal cord [9], brain, and various forms of tumors [67]. In principle, SC cultures can be established from any type of peripheral ganglia, nerve or tissue containing abundant nerve terminals, regardless of anatomical location and developmental stage. Most investigations have been carried out with early postnatal SCs for several reasons. Firstly, the dissociation of immature nerve containing underdeveloped connective tissue layers and unmyelinated fascicles facilitates the digestion with proteolytic enzymes and the recovery of highly viable mother SCs. Secondly, postnatal SCs proliferate at a fast rate and can be maintained for over 100 consecutive passages without displaying evidence of replicative senescence, transformation or aberrant growth [68]. Albeit the culturing of SCs from mature (adult) nerves is more labor intensive, special methods are available to efficiently procure myelin-free, purified, SCs from adult humans and many animal species [69]. With the proper biosafety measures, optimization of the dissection technique and other culture variables, large numbers of SCs can be retrieved from a relatively small segment of an adult human nerve [70]. As mentioned previously, the sural nerve has been preferred for human SC isolation due to the ease of access and its procurement through a minimally invasive procedure that does not compromise other bodily functions. In experimental animals, SCs are prevalently obtained from the sciatic nerve to maximize initial cell yields per mass of tissue. SCPs can be prepared from enzymatic digestion of developing nerves dissected from rat and mouse embryos at day 14 and day 12 of gestation, respectively [71]. Alternative sources of cultured SCs are the DRG from embryonic, postnatal and adult animals, the spiral (cochlear) ganglia [72], the nerve roots, and the skin [73,74]. 

### 3.2. Isolation and Management of the Initial Populations

Lineage-committed SCs survive well after being extracted from the nerve environment due to the developmental acquisition of autocrine survival circuits [75]. When SCs are isolated at the precursor (embryonic) stage, they require provision of neuregulin in the culture medium as surrogate for endogenous survival factors expressed by axons [76]. SCPs differentiate into SCs on schedule in neuregulin-containing medium and their progeny becomes independent of added prosurvival factors [77]. However, it should be noted that all normal, nerve-derived SCs require exogenously provided mitogenic factors (or live axons) for expansion in vitro. Though cultured (normal) SCs secrete neuregulin [78] and other potentially mitogenic factors [77], autocrine factors are normally not sufficient to promote cell division in culture. 

SC cultures were initially obtained from the axonal outgrowth emanating from DRG explants (Section 2) but these preparations not only contained a small number of SCs but also assorted contaminating phenotypes. A more efficient recovery of SCs can be achieved from the proteolytic disintegration of the sciatic nerve of animals at an early postnatal stage, i.e., postnatal days 1 to 3 in rats, immediately after harvesting [50], or alternatively, from the enzymatic dissociation of mature nerves preferably after a period of in vitro predegeneration [79]. Adult nerves render higher numbers of mother SCs per unit/animal but the dissociation of the nerves needs to be optimized in order to preserve SC viability while removing the thick myelin sheaths and the compact ECM. A harvest of clean endoneurial fascicles can be achieved by performing a careful dissection of the nerves to remove as much connective tissue as possible [70,80]; this step is crucial to minimize fibroblast contamination in the final SC cultures.

Isolated SCs require prompt anchorage to a substrate immediately after isolation because their survival can be compromised by apoptotic cell death in response to deprivation of adhesion signals (anoikis). In fact, SCs would not survive, proliferate, or differentiate effectively if adhesion, or trophic support, is suboptimal. Substrate requirements vary substantially among distinct SC populations but generally, normal SCs do not adhere well to culture-treated plastic dishes. Culture surfaces need to be coated with poly-lysine or an equivalent substrate at a minimum but most SC populations are best propagated on plates coated with collagen or laminin, two natural components of the nerve’s ECM. Nevertheless, it is recommended to empirically test the adhesion properties of each SC population, as this property is affected by environmental factors and the state of maturation of the cells. It is nowadays possible to tailor not only the chemical composition but also the physical properties of the substrate, such as the stiffness or elasticity, to change the adhesion properties of SCs and their behavior in vitro [81,82,83].

As mentioned above, postnatal nerves can be dissociated after a short period of incubation with a trypsin-containing solution supplemented with a small concentration of collagenase to render SCs ready to plate within the same day. However, this is not possible with adult nerves. To isolate adult nerve SCs, traditional methods have used predegeneration of nerve fibers in culture for at least 1-2 weeks with or without multiple explantation of adherent tissue fragments [70]. It has been shown that when endogenous SCs are allowed to digest myelin debris and actively proliferate within floating or adherent degenerating nerve fragments, the recovery of viable cells is greatly enhanced [70,79]. Predegeneration and explantation of the nerve segments from and into serial plastic surfaces is a practice that facilitates the migration and removal of contaminating fibroblasts [79]. Nonetheless, it is possible to safely isolate adult SCs immediately after nerve harvesting typically by using a combination of gentle mechanical teasing and chemical dissociation [67,80]. 

### 3.3. Amplification and Subculture 

Most SC cultures propagate for several passages provided the substrate and extracellular factors are continuously renewed. Postnatal rat SCs can be expanded nearly indefinitely without undergoing senescence [68] but lack of replicative senescence is not a common feature of cultured SCs. SCs from humans, monkeys, pigs, dogs, and mice have a limited lifespan in vitro [69]. For instance, adult human SCs expand steadily for up to 6 consecutive passages before they reach a state in which no further expansion occurs possibly due to senescence. Even so, by using a dilution factor of 10-fold with each passage, an overall expansion of over 10^5^ times the original number of starting cells can be obtained [84]. This scheme of amplification has allowed the generation and testing of autologous human SCs in clinical cell therapy [17]. Embryonic and early postnatal SCs naturally exhibit a higher proliferation rate than the adult counterparts but adult nerve-derived SCs are highly proliferative in mitogen-supplemented media. For instance, we have shown that dissociation of adult rat sciatic nerve tissue can yield >10^8^ cells/nerve at passage-1 with cells duplicating on average every 2–3 days [80]. 

Cultured SCs are sensitive to a range of growth factors other than neuregulin. These factors include platelet-derived growth factor, insulin-like growth factor, fibroblast growth factor, transforming growth factor [85], hepatocyte growth factor [86], calcitonin gene-related peptide [87], Gas6 [88], and adenosine [89]. Yet, neuregulin seems to exert the most potent mitogenic action across a wide range of species possibly due to its well described pleiotropic effects on proliferation, survival, metabolic activity and migration [90]. Purified heregulin-β1 has been used broadly in culture media formulations. The availability of a soluble recombinant neuregulin peptide comprising the highly conserved EGF-like domain of neuregulin-β1, instead of the whole protein, has provided a cost-effective way to supplement culture media intended for large and clinical grade preparations of SCs [17]. This peptide is sufficient to elicit strong and sustained ErbB2-ErbB3 activation in SCs when provided in combination with cAMP-inducing agents such as forskolin (a direct activator of the adenylyl cyclase), [91]. There is no or poor cAMP-inducing activity in undefined components like pituitary extract and serum [37], or the neuregulin protein itself. Although the combination of forskolin and cholera toxin was found to achieve synergistic induction of cAMP in human SCs [53], the former is preferred due to its minimal cytotoxicity and reversible action on the adenylyl cyclase. It should be mentioned, however, that cAMP-inducing agents are not sufficient on their own to drive proliferation of isolated SCs and that cAMP’s effects on SCs are also highly dependent on the dose. Whereas cAMP is a co-mitogenic factor when provided in the lower concentration range, it induces cell cycle arrest, myelin gene expression and differentiation at higher doses [92] and its effects may vary as a function of the species or other factors [69]. Serum does not contain a strong mitogenic activity for SCs and it is not required for SC proliferation induced by a variety of growth factors [38]. However, it is regularly provided in the culture medium to maintain SC homeostasis and survival, and to allow sustained multiplication of the populations over time. 

### 3.4. Phenotypic Identification and Purification 

Nerve-derived SC cultures obtained by traditional methods usually consist of two main types of cells, i.e., SCs derived from the fascicles and fibroblasts from the connective tissue layers, as well as from within the nerve fascicles. Cultured rodent SCs are rather constant regarding their shape, size and levels of expression of SC-specific markers [50,93]. This has facilitated the identification of phenotypes by immunological methods and the elimination of contaminating cells. Immunodetection of S100β can be used alone or together with other markers to confirm the SC phenotype and readily discern SCs from non-glial cell types such as fibroblasts, blood cells (macrophages), endothelial cells and stem cells. One of such markers is the low-affinity neurotrophin receptor or NGFR (the Ran-1 antigen), a neural crest marker which is expressed in SCs throughout development with the exception of the mature, myelinating phenotype. Fibroblasts rather than SCs express vast amounts of fibronectin and Thy-1/CD90, which is a membrane-anchored glycoprotein traditionally used as preferred fibroblast marker in rat SC cultures [94]. The expression of ErbB3 rather than ErbB2 denotes lineage-specificity in the neural crest. Detection of ErbB3 can be useful to identify SCs because its expression does not decline with maturation. Myelin protein zero or MPZ, which is the major structural PNS myelin glycoprotein, is constitutively and specifically expressed in SC cultures [95] but the levels are a minimal expression of those found in myelinating SCs [96]. Other faithful SC identifiers are Sox10, PMP22, L1CAM, and GFAP, with a caveat that the latter marker is poorly expressed in cultures of human origin [67]. 

Thus far, most protocols have been optimized from the outset to render SC cultures nearly devoid of fibroblasts. Mechanical elimination of the epineurium highly reduces the impact of contaminating cells. In addition, media formulations that contain cAMP-inducing agents provide a selective advantage to SCs because cAMP accelerates the rate of SC expansion while concurrently attenuating the propagation of fibroblasts. In spite of that, overgrowth of non-glial cell types can happen at any time. Multiple chemical and physical methods that exploit the differential biological properties of SCs and fibroblasts can be used to increase SC purity, including: (1) the addition of antimitotic drugs such as cytosine arabinoside [50] and the anti-nutrient D-valine to the culture medium [97], and (2) the use of differential adhesion (or release) to a plastic substrate with or without a temperature shock [67,98,99]. Reliable immunoselection methods to discriminate and readily eliminate non-glial cells are complement-mediated cell lysis [50], immunopanning [100], and magnetic- or fluorescent-activated cell sorting [74,80,101]. 

### 3.5. Labeling, Tracing and Gene Delivery

SCs can be infected with viruses [15,102] and transfected at high efficiency [72,99] to express a transgene of interest. They can also be selectively targeted by liposomal formulations to deliver therapeutic nanoparticles [103]. Different non-viral and viral vectors encoding gene reporters have been used to facilitate the tracing and analysis of SCs in vitro and in transplanted animals. For instance, SCs have been genetically modified to ectopically express one or more potentially therapeutic genes to enhance their ability to promote regeneration. Examples include neurotrophic factors [104], cell adhesion molecules [105], and glial scar degrading enzymes [102]. These and other manipulations are possible due to the high metabolic activity, potential for expansion, and maintenance of phenotype of SCs in vitro (Section 4.1). 

### 3.6. Cell Banking

Long-term preservation of SCs can be performed following standard protocols for cryogenic storage of cell lines. With the proper handling, SCs subjected to cryopreservation can be recovered as biologically competent cells with negligible loss of viability [80]. Although the clinical relevance of this step may be questionable, cryopreservation has many advantages for the transfer and sharing of cells, delayed analysis, and comparison of cells derived from independent isolation experiments, batches or donors (Figure 3). Cryogenic storage of human SCs is feasible and safe, for it does not reduce the rate of growth or the capacity to promote regeneration after grafting in the contused spinal cord [84].

## 4. Attributes of Primary and Expanded SCs

As explained in previous sections, cultured SCs maintain their dependency on extracellular signals for several passages. They usually survive for prolonged periods of time but remain quiescent unless they are stimulated with axons or soluble mitogens. Given the right conditions, SCs can achieve full differentiation in vitro by following a process that mimics developmental myelination [20]. They are also capable of getting reactive, as manifested by their capacity to engulf myelin and target it for degradation at the time of undergoing proliferation, in a pattern reminiscent of Wallerian degeneration [25] without the aid of macrophages or other supportive cells [106]. In sum, plasticity with preservation of lineage commitment is perhaps the most remarkable attribute of cultured SCs and one that is differential with CNS glial cells. This attribute explains the exceptional versatility of cultured SCs to proliferate (Section 4.1), differentiate (Section 4.2) and foster axon growth (Section 4.3) when maintained in vitro or re-introduced in the body. 

### 4.1. Maintenance of Phenotype and Proliferation Controls

With the exception of SCPs which maintain some multipotency in vitro [3], SCs from mature and immature nerves are committed and cannot convert into a different phenotype under standard growth conditions. Though significant plasticity is possible, phenotypic transformations are restricted to developmental stages within the SC lineage. The proliferation of cultured SCs is density-dependent [107] and the cells are particularly resistant to transform spontaneously and upon induced oncogenesis. Spontaneous transformation of cultured SCs has been reported but it is a rare occurrence. Multiple genetic changes, such as the concurrent expression of membrane and nuclear oncogenes, are needed for anchorage-independent SC proliferation [108] as evidenced by the observation that overexpression of the strong oncogenic form of the Ras gene, v-Ras, in SCs induces cell cycle arrest rather than transformation. Indefinite in vitro expansion of postnatal rat SCs without loss of anchorage-dependency is possible [68,109,110]. Yet, excessive passaging (>10 passages) of SCs is not recommended as a general practice because genetic changes may accumulate over time [110] and autocrine growth inhibiting factors may be lost [109]. Late passage SCs are mostly undistinguishable from the early passage SCs; they can ensheath axons but not form a myelin sheath, suggesting that the cells maintain their lineage specificity but their capacity to undergo full redifferentation may decline with extensive subculture [111]. Even though immortalized rat SCs can lead to tumor formation after injection in the nerve of adult rats [111], it should be noted that this has not been observed with mitogen-expanded human SCs, which do not give rise to tumors when transplanted in the spinal cord and the sciatic nerve of rats and mice [66,84]. 

### 4.2. Potential for Differentiation 

It was earlier observed that rat SCs placed in culture no longer expressed detectable amounts of myelin-related molecules such as galactocerebroside (GALC), sulfatide (sulfated GALC), MPZ and myelin basic protein (MBP) despite their association with neurites in short-term explant DRG cultures [31]. For instance, the expression of the membrane-associated antigen recognized by the monoclonal antibody O4 (sulfatide) in isolated SCs is not autonomous as seen in cultured oligodendrocytes but dependent on axon contact or the exogenous provision of cAMP-inducing agents [112]. SCs of postnatal and adult origin become O4 negative as soon as they are being removed from the nerves [31]. Not only do SCs lose myelin-gene expression in vitro but can endlessly remain cycling unless exposed to strong instructive signals for differentiation. This attribute stands in stark contrast to that of cultured oligodendrocytes, which spontaneously turn on a myelin gene expression program in the absence of neurons and without experimental induction. 

Whereas SCs return to an immature state after denervation, they are still able to re-differentiate and form myelin in culture and in vivo under appropriate conditions. In co-culture with neuronal cells, axon-related SCs can be induced to form compact myelin by supplementing the culture medium with L-ascorbic acid preferably in the presence of serum [43]. In the absence of neurons, however, prolonged and sustained stimulation with cell permeable analogs of cAMP is sufficient to drive and maintain high levels of expression of proteins and lipids that are specific to myelinating cells but no myelin differentiation occurs and changes are reversible upon the elimination of cAMP-inducing agents [113]. Examples of typical cAMP-inducible genes include transcription factors responsible for the initiation of the myelin program such as Krox-20/Egr-2 and SCIP [114], myelin-associated proteins such as MPZ, periaxin (PRX), and myelin-associated glycoprotein (MAG), [112,115], myelin-specific lipids such as GALC [116], and enzymes responsible for myelin lipid metabolism. To conclude, cultured SCs maintain a stable phenotype and low levels of myelin genes despite establishing contact with axons if this occurs in defined medium [42] but the culture conditions can be manipulated to concurrently promote cell cycle arrest and differentiation both in the absence and presence of neuronal cells. 

### 4.3. Promotion of Neuronal Health and Axon Growth 

Early research established that SCs are a source of survival and trophic factors for neurons and that their neuron-supportive features persist in vitro [117]. Cultured SCs exhibit the expression of a secretory profile including various neurotrophic factors along with cell surface adhesion and ECM molecules that foster axon guidance, elongation and maintenance. This attribute of the cultured SC along with their potential for amplification, has been exploited to use them in cell therapy to promote nerve regeneration (Section 5.1). The observation that SC grafts that are implanted in the lesioned spinal cord can promote the growth of both sensory and propiospinal axons [118] provided an early indication of a generic mechanism of SC-mediated axon support and one that is not restricted to neurons in the PNS. However, the mechanism by which SCs enhance axon extension is complex. It was found recently that cultured SCs release particulate products in the form of extracellular vesicles or exosomes that are internalized by neurons. SC exosomes are stage-specific and promote axon regeneration in culture and after sciatic nerve injury [119]. 

## 5. SC cultures in Translational Research

The applications of cultured SCs are broad. For simplicity, this section has been focused on introducing two key areas with high potential for translation: cellular therapies (Section 5.1) and in vitro modeling of normal and abnormal SC development and physiology (Section 5.2). 

### 5.1. Cell Therapy

Many preclinical studies have shown that SCs obtained via in vitro culture methods form a bridge across a lesion site, recruit endogenous SCs, promote axon growth and sparing, and provide a new myelin sheath to regenerated and demyelinated axons in the PNS [16,120,121] and CNS [10,14,118]. SCs do not naturally reside in the CNS but cells with SC characteristics can be found within traumatic and demyelinating lesions in the brain and the spinal cord [122]. Grafted SCs remyelinate central axons and can be mass-produced for autotransplantation therapy in the CNS, which make them a better suited candidate for myelin repair than the hard-to-access oligodendrocyte precursor cell. 

The prospect of implanting cultured SCs to correct the deficiencies of an injury or disease was anticipated by Richard Bunge as early as in 1975 [18] when methodologies to obtain neuron-free SC cultures were not yet available. Nevertheless, clinical translation of human SC transplants was accomplished only recently. Whereas no major safety concerns have been raised so far, the efficacy of SC transplantation in the brain, spinal cord and nerve remains to be confirmed [recently reviewed in [8,11]).

### 5.2. In Vitro Modeling

Cultured SCs are irreplaceable tools for modeling human diseases considering the multiplicity of conditions that affect SC function. SC cultures have been used from the onset to understand mechanisms of disease in patient-specific cells [123]. The possibility to screen and test potentially therapeutic genes and drugs in SC cultures has immense value to finding relevant treatments for acquired and inherited demyelinating neuropathies [124], familiar forms of cancer, and infectious diseases. Insights into the mechanisms of cancer progression and therapeutic control have been possible from studying cultured SCs from mice and humans, as the SC is the cell of origin on at least two distinctive forms of peripheral nerve tumors arising from mutations in genes involved in cell cycle control [125]. The impact of dysregulated SC metabolism as a result of diabetes, treatment with chemotherapeutic drugs and the presence of circulating auto-antibodies, all of them associated with induced peripheral neuropathies in humans, have benefited from investigations carried out using cultured SCs [126]. Moreover, the clarification of the mechanism of infection and reprogramming by *Mycobacterium leprae*, the causative agent of leprosy, was made possible from studying bacteria-SC interactions in culture [127]. In regards to trauma, the study of axotomy in SC–neuron cultures in compartmentalized chambers provided basic insights into the molecular mechanism of injury-induced demyelination [128]. Modeling SC–astrocyte boundaries in vitro as a means to mimic SC interactions with the glial scar can be used to study interventions to increase axon regeneration by SC grafts [129]. These and many other examples highlight the versatile translational opportunities offered by in vitro systems that consist of—or contain—cultured SCs. 

## 6. Long-Standing Assumptions in SC Culture and Their Implications 

The sections below aim to challenge some concepts in SC biology and culturing that have dominated our thinking for decades in an attempt to highlight gaps in our knowledge and discuss perspectives for future research. Some of the questions that are delineated here have emerged from our direct observations and troubleshooting efforts in managing SCs from nerve tissues and using them in in vitro models of SC differentiation. 

### 6.1. Cultured SCs Are Spindle-Shaped Cells That Express the Marker S100β 

Cultured SCs usually look fairly similar under the phase contrast microscope. They are most often elongated and extend long membranous processes to opposite sides of the cells, thus giving the impression of a bipolar shape. When established on two-dimensional substrates, they tend to form swirling patterns reminiscent of a fingerprint at confluency. Experimenters have often assumed that morphological features are useful for SC identification and discrimination from contaminating cells. However, the size and shape of the cultured SCs are highly variable and susceptible to environmental conditions. SCs (rat) in culture can grow (i.e., increase their size) steadily in serum-containing medium, and even undergo cell division, after having increased their size over 100-fold [130]. Growth arrested cells that are senescent or differentiated can also misguide the eye of the experimenter due to their fibroblastic shape [116]. Thus, it is advisable to make use of different tests to confirm the SC phenotype and not rely solely on the shape of the cells or the detection of markers such as S100β. Immunodetection of SC-specific markers is key but high-resolution forms of cell identification such as transcriptome profiling or proteomics [67,121,131] are most helpful to either confirm or disprove the origin and stage of differentiation of the cells. Although the expression of S100β can be maintained constant, some differences emerge when deep RNA sequencing is performed [132] or the cells are investigated for their levels of expression of growth factors [133]. Positiveness for S100β is suggestive of the SC phenotype but using this intracellular marker as primary measure of SC identity is only informative when the cultures are unmistakably derived from normal peripheral nerve [11]. The use of more exhaustive forms of phenotypic validation is highly recommended when unconventional tissues or cells (e.g., non-neural stem cells) are used to derive the SC cultures or contamination with CNS tissue cannot be ruled out. Unintended introduction of CNS glia can occur when the SC cultures are derived from tissues with close anatomical proximity to the CNS, such as in the case of nerve roots, DRGs, and cauda equine. A minimal CNS-derived fragment can inadvertently introduce immature oligodendrocytes and/or S100β positive, GFAP positive astrocytes. For this reason, detection of SC- or neural crest-specific markers, such as NGFR and ErbB3, can help avoid false positives in cultures suspected of CNS glia contamination. 

### 6.2. Cultured SCs Can Myelinate Axons

Achieving myelin formation is the most desirable endpoint for cultured SCs in general as it represents the highest level of maturation. Whereas oligodendrocytes can form myelin-like membranes in the absence of neurons and even provide a myelin sheath to artificial nanofibers [134], SCs would only form myelin in response to signals emanating from living axons. Besides, myelination is seen only when SCs have succeeded in sorting the axons into one-to-one SC–axon units and when the axons have achieved a minimum diameter of about 1 micron. To date, most studies have employed dissociated or non-dissociated DRG neurons whose neurites provide a substrate for SCs to attach, proliferate, and in due time differentiate (Figure 2). It is common practice to purify the SC cultures as well as to subject them to expansion before seeding them onto the neurons to demonstrate the SC’s functionality towards myelin differentiation [43]. The availability of neuronal cells rather than SCs poses a limit to large-scale experimentation but empirical data have shown that not all axonal systems are myelinatable by SCs in vitro. Even though SCs can myelinate different types of axons in vivo, including axons of CNS origin, sensory neurons from the DRG are the most popular for SC myelination in vitro. To our knowledge, DRG neurons constitute the only well-documented system supportive of myelination after neuron–SC recombination in culture with the consideration that not all individual DRG axons are effectively myelinated by SCs. One clear example is the observation that the DRG axons themselves partition into those that support MPZ expression and myelin formation by axon-engaged SCs and those that do not [135]. The receptivity of axons for myelination can vary according to NGF-dependent signals [136], the levels of expression of neuregulin I type III [137], and possibly other factors. 

Another consideration resultant from empirical observations is that not all preparations of SCs myelinate axons to the same extent. For instance, SCs obtained from human nerves cannot myelinate DRG axons under conditions that allow myelin formation by rat SCs [132,138]. Although the organotypic mouse DRG explants can sustain myelination [139], reconstitution of the neuron–SC relationship after purification of SCs from mice nerves is often problematic. It is noteworthy that DRG neurons from rats and humans are effectively myelinated by rat SCs either if the neurons are primary [138] or derived from iPSCs, [140] but human SCs fail to myelinate DRG axons of both human or rat origin [138]. Myelination by SCs obtained from embryonic, adult or postnatal rat nerves or ganglia has been reported but the efficiency of myelination varies according to experimental variables that affect the neuronal and the SC components both independently and together. Also worth mentioning is that only a minor proportion of the SC population myelinate axons in vitro even under optimal conditions. Most SCs that associate to apparently healthy axons do not transition into an O1 (GALC) positive state despite prolonged co-culture or concurrent provision of soluble enhancers of myelination such as L-ascorbic acid and cAMP agonists [48]. Patchiness or uneven distribution of myelinating SCs within the culture network is a common phenomenon that can misguide the experimenter’s impression on the efficiency of myelination. Even more labor intensive is assessing whether SCs have differentiated into the mature non-myelinating phenotype as it occurs in Remak fibers in vivo. The presence of Remak SCs cannot be appreciated in vitro without the aid of TEM. 

For over 30 years, we have assumed that cells identifiable as SCs in a culture dish are capable of myelinogenesis. We have also assumed that SCs from different origins and organisms would myelinate axons under conditions that are appropriate for rat SCs. How much expansion and purification affect the myelination rate of SCs in culture is mostly unknown but there is evidence indicating that overly expanded rat SCs tend to lose their efficiency to myelinate [111]. Whether the redifferentiation potential is dictated by the cell of origin, the developmental stage, the history of the cells in vitro, or other factors merits a particular assessment. SCs may be a larger group of closely related phenotypes than initially assumed with variants that reflect their origin and function in the motor and sensory nerves [141]. Cell type-specific features that are attributable to the prior residence of the SCs in proximal or distal parts of the nerves may also affect the properties of the cells that are derived in the dish [133]. 

A final point worth recognizing is that of the two mature SC phenotypes found in the nerves, i.e., myelinating and Remak SCs, only the former has been consistently observed in reconstituted SC–neuron models. Structures reminiscent to Remak SC ensheathment were revealed by TEM observations in complex cultures of rat superior cervical ganglion neurons containing fibroblasts [142]. Why the late-developing Remak SC phenotype is inefficiently reproduced in vitro is not known. 

### 6.3. Soluble Mitogens Are Axon-Mimetics That Maintain the SC Phenotype In Vitro

Historically, SC cultures transitioned from complex organotypic systems to simplified SC cultures devoid of neurons (Section 2). This simplification required the introduction of two-dimensional substrates for SC anchorage and soluble chemicals to mimic axonal growth factors but undefined components such as serum, pituitary extract, embryo extract, or complex supplements are also often included. Whereas the advantages of neuron-free cultures are obvious, how much these added factors and substrates mimic the signals that emanate from axons and the nerve’s ECM, respectively, is uncertain. Though axons express neuregulin, and neuregulin elicits SC proliferation, axon-derived factors are complex and their activity on SCs may not be restricted to the induction of ErbB signaling [47]. Besides, SCs exhibit durotaxis and are able to transduce mechanical signals via multiple molecular mechanisms [2]. In recent years, researchers have explored the use of biomaterials with altered chemical, topographical and mechanical characteristics to more closely mimic the properties of peripheral nerve tissues to direct cell fate and promote regeneration [143]. 

Axon deprivation in vitro, with or without expansion, has been generally regarded to minimally affect SC function based on early studies showing that expanded rat SC populations maintained myelinating capability [144]. Still, the genetic and epigenetic changes that the artificially recreated environment can introduce in the nerve-deprived SCs or their progeny are largely unknown. Despite the retention of a lineage-specific molecular phenotype, the cultured SCs may either lose potential to redifferentiate or acquire premature senescence. Coincidentally, the experimental rat is the only reported species in which efficient myelin formation in vitro and lack of replicative senescence with expansion have been documented. Whether the inefficient redifferentiation of some isolated SC preparations is spontaneous or induced, permanent or reversible, or linked to environmental stressors is not known. What seems evident is that restitution of axon contact after expansion as single cells seems insufficient for restoration of human SC differentiation in vitro [132] and in vivo [84].

### 6.4. Any Type of SC Can Give Rise to a Primary Culture 

In general, we have understood, based on experimental data, that the tissue used to originate a primary SC culture may not be restricted to a given developmental stage or sort. Cultured SCs are proliferative and express reasonably high levels of genes that are most typically found in neural crest cells (e.g., ErbB3, NGFR, N-cadherin, Nestin, Vimentin, Sox2, and NCAM1) and SCPs (e.g., Cadherin-19, PLP1, and PMP22) [3,4]. This phenotype is rather invariant irrespective of the type of nerve used to generate the culture, the developmental stage or age of the donor, the species or even the method of isolation. Nearly identical human and rat SC cultures are obtained if the adult nerves are processed by following immediate or delayed dissociation [67,70,79,80]. Most notably, skin-derived human SCs are transcriptionally undistinguishable from the nerve-derived SCs [73]. On the basis of cell surface markers and transcriptomics analysis, Etxaniz et al. (2014) argued that the neural precursor cells in the human dermis entail a SC-related phenotype and that these precursors can be found broadly in relation to the abundance of SCs within all tissues and their capacity to be activated upon damage [74].

The existence of a remnant pool of undifferentiated SCs or SCPs in adult nerves seems elusive. Therefore, it has been generally assumed that SC cultures derive from dedifferentiating (activated) SCs (myelinating and/or Remak) that have acquired a proliferative phenotype after denervation. On one hand, we have assumed that all SCs in the nerves have potential to be isolated as viable, adherent cells able to become part of the primary culture. On the other hand, we have assumed that cells in culture are equally potent to proliferate and re-differentiate. Studies have indicated that SCs retain some memory of their origin in the nerve at least for some time while maintained in culture [133]. Whether the dissimilar capabilities of individual SC populations (Section 6.2) are due to differential origin of the mother cells or distinct lineage commitment prior to or after culturing is unclear. Understanding the cell of origin and its fate in vitro by tracing the cells from the nerve to the culture would prove useful to answer these and other questions. 

## 7. Conclusions and Perspectives

This review intended to present the many advantageous features of SC cultures and the well-established protocols used to derive them to introduce the assorted models and technologies available to date. It also intended to summarize the progress made since the onset of investigations over 60 years ago to illustrate the strict interrelationship between our understanding of SCs in vitro and in vivo, and the development of methods for SC culturing. The historical perspective is important to realize that many of our modern ideas on SC development and myelination were derived from pioneering studies using cultured tissues and cells. In fact, these cultures were the only available physiologically relevant system in which the SC phenotype could be targeted specifically, and experimentally manipulated, up until animal models with genetically altered genes in SCs were developed in mammalian and non-mammalian species. Understanding the evolution of ideas is important when attempting to translate the knowledge acquired from studying rodent cells to humans. Because rodents (mostly rats) have been prevalently used for SC culturing (Section 2), the knowledge gained from exploring cultured rat SCs has been generalized conceptually in the literature. Weighing against this approach, empiric data have revealed that SCs obtained from humans and large animal models differ substantially from those of rats even when similar methods are used to prepare them (discussed in [11]). An understanding of the advantages and limitations of SCs in vitro is key for therapeutic development of tissues and cells, and the range of technologies that have integrated SC cultures, alone or together with other elements (e.g., other cells, scaffolds and biomaterials), to create new models for research or grafts for implantation. 

The derivation of SCs from unconventional phenotypes is an alternative route to generate SCs in a dish. Though this approach may not be necessary for rat SCs due to the high amplification potential and stability of the cells, it can certainly resolve the problem of human SC availability and the ethical concerns that are usually associated regarding nerve tissue procurement. Exploiting the complementary, non-overlapping value of traditional and emerging approaches is essential. Because SCs from cognate PNS tissues more closely mimic those that are endogenous, they have incremental value over engineered cells and cell lines albeit the limitations expressed above. SC isolation directly from PNS tissues is at present the only viable option for clinical cell therapy but it is expected that advances in iSC derivation will soon allow mass production of quality grade, donor-specific SCs without the need to surgically remove a patient’s nerve.

Clearly, standard methodologies for SC culture are both practical and transferable but important challenges lie ahead to improve the quality, stability, and biological activity of natural and engineered SC products. Lastly, considering the versatility and robustness of SC cultures, it is highly likely that they will help advance research on SC biology and therapeutics for many years to come.

## Figures and Tables

**Figure 1 cells-09-01848-f001:**
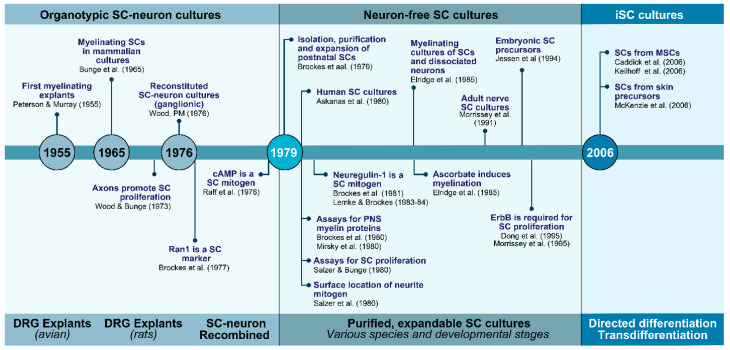
The evolution of SC cultures: from tissue culture to cell culture and beyond. The timeline summarizes some of the major technological and conceptual achievements by many contributing laboratories for over half a century. Main advancements facilitated the transition from culturing SCs within neural tissues (1960s and 1970s) to culturing SCs from neural tissues but in the absence of other cell types (1970s and 1980s). A new set of in vitro technologies are behind the creation of SCs from non-neural cells such as MSCs (2000s-). Some key innovations were represented in the timeline itself along with their references. Other descriptions and citations can be found in the text.

**Figure 2 cells-09-01848-f002:**
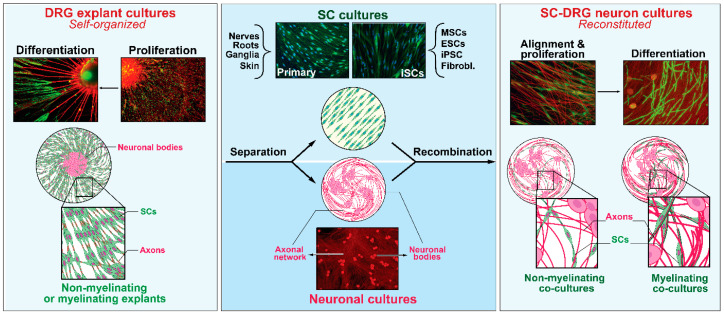
The plasticity of SCs in vitro: deconstruction and reconstitution of neural systems. Explant cultures of sensory (DRG) ganglia (**left**) are complex self-assembled culture systems that rely on the natural regenerative ability of sensory neurons to extend neurites on a two-dimensional substrate while concomitantly endowing endogenous SCs with an opportunity to engage, proliferate and differentiate as it occurs during developmental nerve growth and maturation (**left panel**). The neural and SC components from the ganglia itself or other sources (e.g., nerves, roots, and stem cells) can be recombined to create a simpler cellular system supportive of SC–axon engagement and maturation (**right panel**). Further descriptions can be found in the text along with pertinent references. SC cultures and DRG neurons from rats (primary) were prepared as described in [47]. Differentiation protocols and analysis of myelinating cells were described in [48]. The image of iSCs was kindly provided by Dr. Yong Jun Kim [49].

**Figure 3 cells-09-01848-f003:**
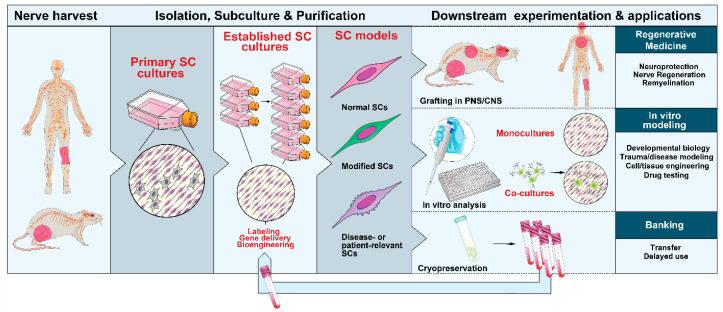
Nerve-derived SC cultures: sources, expandability and applications. SCs can be isolated from neural (PNS) tissues from adult and developing animals and humans. SC cultures can be expanded with the aid of soluble factors and used in a variety of downstream applications, as indicated. Primary or expanded SCs are suitable for diverse manipulations. They can be induced to acquire mature characteristics both in vitro (with and without neurons) and in vivo after transplantation in the PNS or CNS.

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
