# Peer review of "Schwann Cell Cultures: Biology, Technology and Therapeutics"

_cells, 2020, doi:10.3390/cells9081848_

Round 1

Reviewer 1 Report

The author has written a very thorough and systematic review of Schwann cell research. An appropriately brief cover of the early history of this field was laid out at the beginning which was then followed by a timely review of current developments and considerations. I appreciate the challenges to current dogmas and operating thoughts. This deserves publication with minimal edits. I have suggested another idea to incorporate below.

In section 6.3 (Soluble mitogens are axon-mimetics that maintain the SC phenotype in vitro), I appreciate this biomimetic commentary and suggest incorporating a few works in the field around culture substrate stiffness/elasticity which is closely linked to the biology of these cells and how they respond in vivo (e.g., durotaxis). A few examples to include or pick from are: 1) https://doi.org/10.3389/fncel.2017.00347 , 2) https://doi.org/10.1016/j.biomaterials.2012.06.006 , 3) doi: 10.1186/s40824-018-0124-z , 4)  https://doi.org/10.1002/jbm.a.36749

Author Response

Many thanks to the Reviewer for the positive comments and suggestions. The recommendation to address the effect of culture substrate/stiffness is greatly appreciated. This is a very important issue which can be found elaborated in Sections 3.2 and 6.3. Added text is demarked in color.

Reviewer 2 Report

The paper "Schwann cell cultures: biology, technology and therapeutics" turns out to be well written, interesting and a good Review of some important aspects of Schwann cell research.

Altogether this is a sound Review, and summarize the progress made in SC culturing, the many advantageous features of SC cultures and the well established protocols used to derive them.

Considering the importance that schwann cells have and will cover in the future, this review represents a good job.

It still needs some minor corrections:

  1. the quality of the images is completely insufficient, a better resolution is necessary.
  2. the abbrevetion should be defined in parentheses the first time they appear, often this does not happen and the full written word is missing.(Line 224,231,232,275,276).
  3. Line 108: move (TEM) after microscopes.
  4. Line 137: delete (of).
  5. Line 411: delete (-) before 1.
  6. Line 513, to correct automonous in autonomous.
  7. Line 555: delete curly bracket.
  8. Line 799, 932: are missing the pages.
  9. Line 867, 891, 908, 911, 916, 957, 1055, 1095: replace the (.) with the (;) before et al.

Author Response

Many thanks to the Reviewer for the positive comments. Essentially all recommendations were followed as suggested with the exception of the formatting of the references (last item in the list) because it resulted from an automated formatting using the required EndNote Style for CELLS. I understand that format changes to the references will be taken care of during production.

The Reviewer may notice that the figures were improved to enlarge most components and make a better use of the space. For this, I used the help of a professional graphic designer. The changes are more noticeable in Figure 1 for its context of text. Unfortunately, the full document seems to convert the figures automatically with a loss in resolution but this is not the case of the original images. I hope the Reviewer can download the figures separately for further scrutiny if the problem of resolution persists. I also used the help of a professional english editor to improve the language and punctuation throughout the text.

Reviewer 3 Report

The manuscript is well written and will be useful to the community. Although the author went through nearly all essential aspects, some important points are not covered yet. For instance, the embryonic origin of a SC lineage can be described in better details as well as the plasticity of SCs and SCPs. It is well known today that SCPs give rise (in vivo!) to the majority of melanocytes, all parasympathetic neurons (plus numerous enteric neurons) and presumably all chromaffin cells of adrenal glands (please see papers from Adameyko, Enomoto, Pachnis and Kalcheim labs). Within the nerves, SCPs give rise to endoneurial fibroblasts (lately re-investigated by Freda Miller lab). The plasticity of SCPs in vitro is also well recognized. It would be great if the author could cover this part for the sake of having this manuscript complete. 

Author Response

Many thanks to the Reviewer for the positive comments. The recommendation to explain the plasticity of Schwann cell precursors (SCPs) is greatly appreciated. This is a very important issue which can be found covered in sections 1 and 4.1. For the purpose of culturing methods, it is important that the readers understand the difference between plasticity regarding lineage specification (SCPs) and plasticity of mature cells (SCs). The culturing of SCPs is referred to also in Section 3.2. I greatly appreciate the recommendation to bring this issue to my attention.    

Round 2

Reviewer 3 Report

I believe the paper is ready to be published. I have no further comments.